# Relationship between Structure and Antibacterial Activity of α-Aminophosphonate Derivatives Obtained via Lipase-Catalyzed Kabachnik−Fields Reaction

**DOI:** 10.3390/ma15113846

**Published:** 2022-05-27

**Authors:** Dominik Koszelewski, Paweł Kowalczyk, Paweł Śmigielski, Jan Samsonowicz-Górski, Karol Kramkowski, Aleksandra Wypych, Mateusz Szymczak, Ryszard Ostaszewski

**Affiliations:** 1Institute of Organic Chemistry PAS, Kasprzaka 44/52, 01-224 Warsaw, Poland; psmigielski98@gmail.com (P.Ś.); jan.samsonowicz-gorski.stud@pw.edu.pl (J.S.-G.); 2Department of Animal Nutrition, The Kielanowski Institute of Animal Physiology and Nutrition, Polish Academy of Sciences, Instytucka 3, 05-110 Jabłonna, Poland; ryszard.ostaszewski@icho.edu.pl; 3Department of Physical Chemistry, Medical University of Bialystok, Kilińskiego 1 Str., 15-089 Białystok, Poland; kkramk@wp.pl; 4Centre for Modern Interdisciplinary Technologies Nicolaus Copernicus University in Torun ul. Wileńska 4, 87-100 Toruń, Poland; wypych@umk.pl; 5Department of Molecular Virology, Institute of Microbiology, Faculty of Biology, University of 7 Warsaw, Miecznikowa 1, 02-096 Warsaw, Poland; mszymczak@biol.uw.edu.pl

**Keywords:** α-aminophosphonates, Kabachnik−Fields reaction, antimicrobial activity

## Abstract

We reported a new method dealing with the synthesis of novel pharmacologically relevant α-aminophosphonate derivatives via a lipase-catalyzed Kabachnik−Fields reaction with yields of up to 93%. The advantages of this protocol are excellent yields, mild reaction conditions, low costs, and sustainability. The developed protocol is applicable to a range of *H*-phosphites and organic amines, providing a wide substrate scope. A new class of α-aminophosphonate analogues possessing P-chiral centers was also synthesized. The synthesized compounds were characterized on the basis of their antimicrobial activities against *E*. *coli*. The impact of the various alkoxy groups on antimicrobial activity was demonstrated. The crucial role of the substituents, located at the aromatic rings in the phenylethyloxy and benzyloxy groups, on the inhibitory action against selected pathogenic *E.* *coli* strains was revealed. The observed results are especially important because of increasing resistance of bacteria to various drugs and antibiotics.

## 1. Introduction

α-Aminophosphonates appear to be a very important class of organic compounds because of their potential biological activities [1,2,3,4]. The distinctive character of bioactive organophosphorus compounds has established their wide applicability in agricultural and medicinal chemistry [5,6,7,8,9,10,11,12]. α-Aminophosphonates (Figure 1) play a crucial role as a platform to design new drugs [13,14,15,16,17,18,19,20]. Among other advantages, there are several reports regarding their antimicrobial activity; alafosfalin, for example, a simple dialkyl α-aminophosphonate, exhibits activity against pathogenic *E. coli*, *S. aureus*, *Bacillus*, and *K. pneumonia* strains (Figure 1) [21,22,23,24,25,26,27,28,29,30]. However, the application of alafosfalin in medicine is limited due to its instability. It is shown that the correct design of the alkoxyl groups in the *H*-phosphite used for these compounds may significantly increase their antimicrobial properties (Figure 1) [1,2,3,31,32,33,34,35,36].

The aim of the work is to develop a metal-free protocol for the preparation of α-aminophosphonate derivatives with P-chiral centers on bacterial strains K12 and R2−R4.

## 2. Materials and Methods

### 2.1. Microorganisms and Media

All microorganisms and media were accurately described in detail in the previous work [37,38,39,40,41,42,43,44,45,46,47,48,49,50,51,52,53,54,55,56,57,58,59,60,61,62,63,64,65,66,67,68,69,70,71,72,73,74] and analyzed by a Tukey test.

### 2.2. General Methods of Synthesis α-Aminophosphonate Derivatives

All the chemicals were described in detail in the previous work [74]. All specific strains, such as *Pseudomonas cepacia* (PcL) and wheat germ, were provided by Sigma-Aldrich (Merck). The bovine acetone powder was prepared in our laboratory according to the literature procedure [38]. Symmetrical and unsymmetrical *H*-phosphites were obtained via alcoholysis of the dimethyl phosphite with the appropriate alcohol according to the literature procedure [40,41,42,43,44,45,46] (see Appendix A).

## 3. Results

### 3.1. Chemistry

Organophosphorus compounds show a variety of relevant biological activities [47,48,49]. The Kabachnik–Fields reaction is the most efficient method for the formation of carbon−phosphorus bonds using an aldehyde, amine, and *H*-phosphite. A number of other synthetic approaches have also been reported for the preparation of α-aminophosphonates [7,50,51,52,53,54,55,56,57,58,59,60,61,62,63,64,65,66,67,68,69,70,71,72,73,74,75,76,77,78,79,80,81,82,83,84,85,86,87,88,89,90,91,92,93,94,95]. These methods are generally conducted in the presence of various organic and inorganic bases [56,57,58] as well as Lewis and Bronsted acids, such as zirconium tetrachloride (ZrCl_4_), aluminum chloride (AlCl_3_), tantalum pentachloride (TaCl_5_), or lanthanide triflates [59,60,61,62,63]. Therefore, it is necessary to further develop an efficient one-pot, multicomponent synthesis of α-aminophosphonates that is devoid of these problems and fulfils requirements of the pharmaceutical industry. Enzymes, which are natural catalysts with high catalytic activity, seem to be the best alternative leading to the development of new synthesis methods that meet the requirements related to safety and environmental protection. In addition, enzymes enable the synthesis of compounds without metal contamination, which is especially appreciated by the pharmaceutical industries. Among other uses, hydrolases are most often used as biocatalysts in organic synthesis. Our work is more focused on discovering new unnatural catalytic activities of hydrolases. This phenomenon was defined as enzymatic promiscuity. Recently, a number of unnatural reactions catalyzed by hydrolases have been reported, such as the aza-Henry reaction [64], Michael additions [65,66], 1,2-addition of thiols to imines [67], and Morita–Baylis–Hillman reaction [68,69,70]. Although some chemical strategies work towards the synthesis of α-aminophosphonates, the biocatalytic preparation of target α-aminophosphonates remains unexploited. It was shown that some selected α-aminophosphonates could be obtained from aniline derivatives by the Kabachnik–Fields reaction using *Candida antarctica* lipase B (CAL-B) as a catalyst [71,72,73].

As a continuation of our research on seeking new catalytic activities of hydrolases [74,75,76,77,78,79,80,94,95], we focused our efforts on elaborating a sustainable metal-free method towards desired α-aminophosphonates **1**–**16** (Figure 2).

Regarding the promiscuous activity of lipases, [71] the model Kabachnik–Fields reaction of benzyl amine (1 mmol), benzaldehyde (1 mmol), and dimethyl phosphite (1 mmol) was conducted in neat at 25 °C (Figure 1 and Table 1, entry 1).

As shown in Table 1, lipase from a porcine pancreas (PpL) was found as the best catalyst among the tested lipases for this addition reaction (Table 1, entry 2). The α-aminophosphonate **1** obtained agood yield (73%) after 24 h in neat at 25 °C. The yield did not increase substantially after 24 h. In the absence of enzyme only traces of the target product **1** was formed (Table 1, entry 1). In addition, four different nonenzymatic catalysts were reported in the literature as sustainable promoters of the Kabachnik–Fields reaction [81,82,83,94,95]; copper(I) iodide, copper(I) oxide, copper(II) acetate, and phenylboronic acid were tested under similar reaction conditions, leading to the target product **1** with up to a 39% yield (Table 1, entries 19–22). It is well recognized that the type of solvent used has a great impact on enzyme stability and activity [84]. Product **1** was provided with the highest yield of 88% in TBME (Table 1, entry 7); therefore, this solvent was applied in the following optimization. Furthermore, the model reaction was carried out at elevated temperatures; however, the yield of product **1** was reduced at temperatures above 30 °C (Table 1, entries 8 and 9). Next, we studied if the amount of enzyme used had any impact on the reaction yield, and we found out that the yield of target compound **1** increased slightly by raising the amount of PpL from 50 mg to 80 mg. Thus, the 80 mg of PpL was the optimal amount for the further investigations [71,94,95].

Finally, we used the elaborated enzymatic protocol with various aromatic and aliphatic amines, aldehydes, and symmetrical as well as unsymmetrical *H*-phosphites [41] (Figure 2). The enzymatic Kabachnik–Fields reaction with aliphatic aldehydes and amines as well as 2-phenylethylamine provided products **4**, **5**, and **9**–**11** with lower yield ranges from 51% to 71% (Figure 2). A similar reduction in the reaction yield was observed for sterically bulky electron-rich aldehyde and amine with methoxy groups located at the phenyl ring, which resulted in product **8** with a 69% yield. Finally, the application of unsymmetrical *H*-phosphonates provided P-chiral products **14**–**16** as a mixture of diasteroisomers (1:1) with yields up to 76%. The structures of all obtained compounds **1**–**16** are presented in the experimental section (Appendix A).

Additional experiments were performed to gather insights on the reaction pathway. Under developed conditions, *N*-(4-methoxylbenzylidene)benzylamine was used together with dimethyl *H*-phosphite in the presence of PpL as a catalyst, which resulted in an excellent yield of 95% of the target α-aminophosphonate **1**. This observation constitutes the initial formation of an imine in the presence of lipase (Figure 2).

### 3.2. Cytotoxic Studies of the Library of α-Hydroxy Phosphonate Derivatives

It is worth noting that the introduction of a fluorine atom into the structure of all 16 tested compounds did not have a significant effect on the activity of **2** and **12**, which is often observed for various types of compounds exhibiting antibacterial activity [12] (Figure 3, Figure 4, Figure 5, Figure 6 and Figure 7).

The analyzed bacterial strains used in the experiments were used in 48-well plates. (Figure 3, Figure 4 and Figure 5 and Table 2).

### 3.3. Analysis of R2–R4 E. coli Strains Modified with α-Aminophosphonate Derivatives

The obtained MIC values as well as our previous studies with various types of the analyzed compounds [85,86,87,88,89,90,91,92,93,94,95] indicate that α-aminophosphonate derivatives also show a strong toxic effect on the analyzed bacterial model strains. The three analyzed compounds were selected for further analysis by modifying their DNA. Modified bacterial DNA was digested with Fpg as described earlier [85,86,87,88,89,90,91,92,93]. All selected analyzed α-aminophosphonate derivatives (Figure 6), including different types of alkoxy groups, substituents located at the phenyl ring, and the length of the alkyl chain, can strongly change the topology of bacterial DNA. After digestion with Fpg, approximately 3.5% of oxidative damage was identified, which very strongly indicates oxidative damage in bacterial DNA, similar to the previous observations [85,86,87,88,89,90,91,92,93]. The different types of alkoxy groups, substituents located at the phenyl ring, and the length of the alkyl affected this outcome (Figure 6).

**Figure 6 materials-15-03846-f006:**
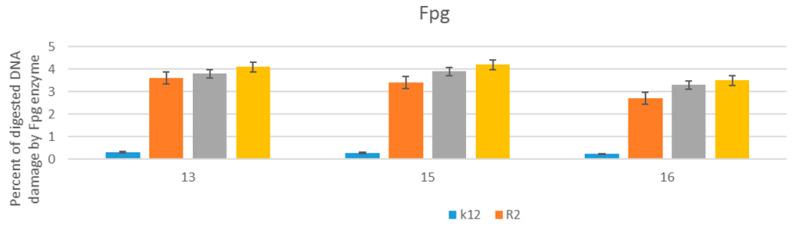
Percentage of plasmid DNA recognized by the Fpg enzymes (*y*-axis) with model bacterial, K12, and R2–R4 strains (*x*-axis). All analyzed compounds numbered were statistically significant at <0.05 (see Table 2).

### 3.4. R2-R4 E. coli Strains with Tested α-Aminophosphonate Derivatives

The performed studies prove that the analyzed and newly synthesized compounds can potentially be used as “substitutes” for the currently used antibiotics in hospital and clinical infections, (Figure 7 and Figure 8 and Appendix A).

**Figure 7 materials-15-03846-f007:**
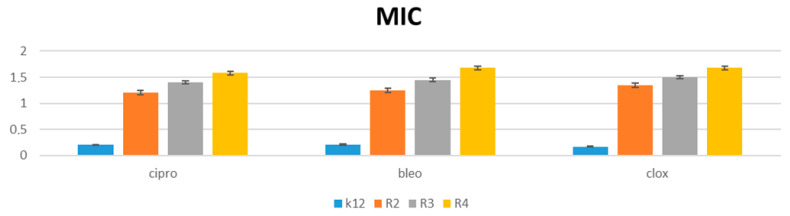
Examples of MIC with model bacterial strains K12, R2, R3, and R4 for studying the antibiotics ciprofloxacin (cipro), bleomycin (bleo), and cloxacillin (clox). The *x*-axis features antibiotics used sequentially. The *y*-axis features the MIC value in µg/mL^−1^.

Large modifications of plasmid DNA were observed for the three analyzed compounds numbered **13**, **15**, and **16**, showing high superselectivity.

## 4. Conclusions

Our developed protocol provides an efficient mild and metal-free synthesis of the target products with a high yield (51–93%). Among the studied derivatives, the compounds possessing alkoxy groups with halogen atoms or nitro groups in phosphate moieties **13**, **15**, and **16** turned out to be the most active compared to derivatives with the dimethyl groups (Figure 2). Finally, the reported α-aminophosphonate derivatives are more cytotoxic in the model bacterial cells than the following commonly used antibiotics: ciprofloxacin, bleomycin, and cloxacillin.

## Data Availability

On request of those interested.

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
