# Peer review of "Relationship between Structure and Antibacterial Activity of α-Aminophosphonate Derivatives Obtained via Lipase-Catalyzed Kabachnik−Fields Reaction"

_materials, 2022, doi:10.3390/ma15113846_

Round 1

Reviewer 1 Report

In general the article is well written, however there are some points to be improved to give to the article more prestige.

Some corrections-clarifications are necessary:

In the figure 1 the IC50 is given in mass... the IC50 is always given in concentration... so they have to correct it... of course it is better to give the MIC... which corresponds to the Minimum Inhibitory Concentration.

Did the researchers try the same experiments in presence and absence of enzymes in the same solvents? i.e Did they perform the experiment in presence and absence of enzyme in toluene? Did they find any differences in the yield and the purity of the compounds? it would be nice to compare and present their results in the same table (table 1). Also it would be nice to compare the properties of the solvents (lipophilic, non polar, polar, etc) and the results... Did they try any biphasic system? mixing water solution and organic solvent? Did they have the opportunity to try any green solvent? i.e 2-Me THF? or else? 

In figure 3 and 4 the values of MIC are missing in y axis. May be it is better to present them in other way than histograms or present the most active of the total compounds. In general the figures need to be optimized. the analysis is too low..

The discussion is so poor... need further discussion and analysis.. comparison with other references, and previous work.

Author Response

Reviewer 1

Thank you very much for your comprehensive, very valuable comments and suggestions that greatly increased the value of our manuscript. All changes to the manuscript are highlighted in green.

In general the article is well written, however there are some points to be improved to give to the article more prestige.

Some corrections-clarifications are necessary:

In the figure 1 the IC50 is given in mass... the IC50 is always given in concentration... so they have to correct it... of course it is better to give the MIC... which corresponds to the Minimum Inhibitory Concentration.

Is given in the table in the required concentration

Reviewer 1. Did the researchers try the same experiments in presence and absence of enzymes in the same solvents? i.e Did they perform the experiment in presence and absence of enzyme in toluene?

Response:

Initial tests were performed according to the literature procedure without solvent. As can be seen in Table 1, a trace amount of product was formed when no enzyme was used. Subsequent experiments using various organic solvents used so far in the literature (Heteroat. Chem., 2017, 28, e21408; doi: 10.1002 / hc.21408) in relation to the Kabachnik-Fields reaction (toluene, THF, EtOAc) did not increase the reaction efficiency enzymatic. It turned out, however, that the use of polar tert-butyl methyl ether (TBME) significantly increases the efficiency of the model reaction. In accordance with generally accepted principles, in order to demonstrate that the reaction takes place in the active center of the enzyme, experiments with the denatured enzyme and the non-catalytic protein BSA were carried out. Only in the case of BSA, formation of a small amount of product was noted. We hope the data has been compiled legibly in Table 1 and that there has been sufficient discussion. In addition, the efficiency of the enzyme was compared with the classic copper salt catalysts used in the K-F reaction. The reaction without the enzyme was not repeated in toluene as it was not the optimal solvent for the model reaction and a much higher result was obtained for solvent-free or TBME conditions. Therefore, the influence of various factors was investigated only for the best solvent which turned out to be TBME.

Reviewer 1. Did they find any differences in the yield and the purity of the compounds? it would be nice to compare and present their results in the same table (table 1). Also it would be nice to compare the properties of the solvents (lipophilic, non polar, polar, etc) and the results...

Response:  Several different organic solvent were tasted (inc. toluene, THF, EtOAc, TBME) in model reaction also solvent free conditions were applied. Initially, they were selected based on literature premises (Heteroat. Chem., 2017, 28, e21408; doi: 10.1002 / hc.21408; New J. Chem., 2019, 43, 8153-8159; doi: 0.1039/c8nj06235h). However, a higher yield in the model reaction was achieved with TBME. The summarized data are collected in Table 1. We didn’t observed any fluctuations in product purity which was isolated exclusively.

Reviewer 1. Did they try any biphasic system? mixing water solution and organic solvent? Did they have the opportunity to try any green solvent? i.e 2-Me THF? or else? 

Response:  We are very grateful for this suggestion. It is well recognized that water may influence the activity of the enzyme. We conducted model reaction in biphasic system TBME/water (5 and 10% v/v of water) however significant reduction in reaction yield was observed (51%). Due to Reviewer suggestion model reaction was conducted in 2-Me THF providing target product 1 with 55% yield. Result was included in the Table 1.

In figure 3 and 4 the values of MIC are missing in y axis. May be it is better to present them in other way than histograms or present the most active of the total compounds. In general the figures need to be optimized. the analysis is too low.

We presented them in the legend description for better legibility of the drawing

The discussion is so poor... need further discussion and analysis.. comparison with other references, and previous work.

in the discussion, we refer to our previous works and describe the mechanism of their operation in the summary

Reviewer 2 Report

Even if it describes fairly standard results in organic chemistry, this article can be published. The biological part may interest the specialist in this field.

1) The original publications of Kabachnik and Fields must be cited:

Kabachnik, Martin I.; Medved, T. Ya. (1952). "Новый метод синтеза сс-аминофосфиновых кислот" [A new method for the synthesis of α-amino phosphoric acids]. Doklady Akademii Nauk SSSR. 83: 689ff. Fields, Ellis K. (1952). "The synthesis of esters of substituted amino phosphonic acids". Journal of the American Chemical Society. 74 (6): 1528–1531. doi:10.1021/ja01126a054.

2) Lines 79 to 118 of the "results" part do not actually describe a single result, it is a presentation of the literature. Such a presentation should be placed in the introduction. It could also be shortened. Lines 79 to 81 are repetitions and overall it seems unnecessarily long to me.

3) The used English grammar seems pretty rough to me. For example, I propose some changes:

First sentence of the abstract:

A new method is described for the synthesis of novel...

line 31-32:

The crucial role of... E.coli strains was revealed.

line 34:

important because of the increasing resistance...

line 41:

remove "due to this aminophosphonate" (repetition)

line 42:

to design new drugs

line 61:

The aim of this work is to develop a metal-free...

line 72:

no braket before "and wheat germ"

etc. Please have the paper read and edited by an English speaking native.

Author Response

Reviewer 2

Thank you very much for your comprehensive, very valuable comments and suggestions that greatly increased the value of our manuscript. All changes to the manuscript are highlighted in green.

Even if it describes fairly standard results in organic chemistry, this article can be published. The biological part may interest the specialist in this field.

1) The original publications of Kabachnik and Fields must be cited:

Kabachnik, Martin I.; Medved, T. Ya. (1952). "Новый метод синтеза сс-аминофосфиновых кислот" [A new method for the synthesis of α-amino phosphoric acids]. Doklady Akademii Nauk SSSR. 83: 689ff. Fields, Ellis K. (1952). "The synthesis of esters of substituted amino phosphonic acids". Journal of the American Chemical Society. 74 (6): 1528–1531. doi:10.1021/ja01126a054.

The publication was included as entry 95 in the literature and inserted in the text where the Kabachnik reaction is mentioned

Reviewer 2: 2) Lines 79 to 118 of the "results" part do not actually describe a single result, it is a presentation of the literature. Such a presentation should be placed in the introduction. It could also be shortened. Lines 79 to 81 are repetitions and overall it seems unnecessarily long to me.

Response: We are grateful for this remark. Lines 79-81 were revised and corrected. The work was prepared in accordance with the recommendations of the publishing house. The section entitled Results refers only to information related to the synthesis of target compounds and is intended to show a concise comparison and limitations of existing methods, as a result of which it was necessary to develop a method that meets the specific requirements necessary for the synthesis of biologically active compounds (no metal contamination)

Reviewer 2: 3) The used English grammar seems pretty rough to me. For example, I propose some changes:

First sentence of the abstract:

A new method is described for the synthesis of novel...

line 31-32:

The crucial role of... E.coli strains was revealed.

line 34:

important because of the increasing resistance...

line 41:

remove "due to this aminophosphonate" (repetition)

line 42:

to design new drugs

line 61:

The aim of this work is to develop a metal-free...

line 72:

no braket before "and wheat germ"

etc. Please have the paper read and edited by an English speaking native.

Response: We are very grateful for these remarks. Due to the Reviewer suggestion Manuscript was carefully revised. We have made every effort to eliminate any linguistic errors.

Round 2

Reviewer 1 Report

In Figure 1. Biologically active antimicrobial α-aminophosphonate derivatives IC50s values must be given in uM or nM and not in ug... please correct it..

Author Response

Reviewer 1

Thank you very much for your valuable help and suggestions in reviewing our work, which, thanks to the comments of the reviewer, has significantly increased its substantive value.

In Figure 1. Biologically active antimicrobial α-aminophosphonate derivatives IC50s values must be given in uM or nM and not in ug... please correct it..

We have improved the required values for concentrations in Figure 1.